# Effect of Tempforming on Strength and Toughness of Medium-Carbon Low-Alloy Steel

**DOI:** 10.3390/ma16031202

**Published:** 2023-01-31

**Authors:** Diana Yuzbekova, Valeriy Dudko, Alexander Pydrin, Sergey Gaidar, Sergey Mironov, Rustam Kaibyshev

**Affiliations:** 1Laboratory of Advanced Steels for Agricultural Machinery, Russian State Agrarian University-Moscow Timiryazev Agricultural Academy, 127550 Moscow, Russia; 2Laboratory of Mechanical Properties of Nanostructured Materials and Superalloys, Belgorod State National Research University, 308015 Belgorod, Russia

**Keywords:** tempforming, ultra-high-strength steel, fracture toughness, mechanical behavior, microstructure, carbides

## Abstract

The effect of tempforming on the strength and fracture toughness of 0.4%C-2%Si-1%Cr-1%Mo-VNb steel was examined. Plate rolling followed by tempering at the same temperature of 600 °C increases yield stress by 25% and the Charpy V-notch impact energy by a factor of ~10. Increasing rolling reduction leads to the reorientation and elongation of grains toward the rolling direction (RD) and the development of a strong {001} <110> (rotated cube) texture component that highly enhances fracture toughness. A lamellar structure with a spacing of 72 nm between boundaries and a lattice dislocation density of ~10^15^ m^−2^ evolves after tempforming at 600 °C with a total strain of 1.4. Two types of delamination were found, attributed to crack branching and the propagation of secondary cracks along the rolling plane perpendicular to the propagation direction of the primary crack. Delamination toughness is associated with the nucleation of secondary cracks in RD and their propagation over a large distance. The critical condition for delamination toughness is the propagation of primary cracks by the ductile fracture mechanism and the propagation of secondary cracks by the brittle quasi-cleavage mechanism.

## 1. Introduction

Low-alloy ultra-high-strength (UHS) steels with yield stress (YS) > 1400 MPa are used in critical components of aerospace vehicles due to a combination of high YS and satisfactory fracture toughness [1]. These steels are subjected to heat treatment consisting of austenitizing followed by water quenching and final low-temperature tempering, which results in a low-temperature-tempered martensitic structure [1,2]. An increase in the Charpy V-notch impact energy is important for achieving the high performance of UHS steels [1]. However, steels may be strong or tough but they are rarely both, since strength and fracture toughness have opposing characteristics [2,3]. These properties could not be improved simultaneously through the modification of conventional heat treatment and the chemical composition of low-alloy steel [1] and, therefore, new approaches must be developed to increase the fracture toughness of UHS steels. High-alloy UHS steels containing ≥9%Co provide a superior combination of strength, ductility, and toughness but the commercial use of these materials is limited by the high cost of the high-performance critical parts used in the aerospace industry.

A method of thermomechanical processing (TMP) called tempforming was proposed by Japanese scientists to increase the toughness at low temperatures of low-alloy steels [3,4,5,6]. This processing enables very high strength and toughness to be achieved in an anisotropic manner for a reasonable cost [3,4,5,6,7,8,9,10]. Tempforming involves austenitizing, water quenching, and tempering at T ≥ 500 °C followed by large strain warm rolling at the same temperature, resulting in the transformation of the tempered martensite lath structure to a lamellar-type structure with a transverse crystallite dimension of ~100 nm and a uniform distribution of dispersed carbides [3,4,5,6,7,8,9,10]. Besides the improvement of impact toughness by a factor of ~10 in the normal direction (ND) to the rolling plane, tempforming may also increase YS [4,8,9]. Y.Kimura et al. [3,4,5,6] showed that a very high YS ≥ 1800 MPa and absorbed impact energy ≥200 J could be achieved in medium-carbon low-alloy steels by tempforming through caliber rolling. Such properties were achieved in a medium-carbon low-alloy steel with the chemical composition shown in Table 1, which considered a prototype for the development of new UHS steel for high-performance bolts [4]. It is worth noting that the formation of the lamellar structure induces crack-arrester-type delamination, in which the propagation of the main crack in the transverse direction (TD)–ND plane leads to the branching and propagation of secondary cracks along the rolling direction (RD)–TD plane [4,7,9]. The tip of the main crack is blunted and provides high-impact toughness [4,7,9]. However, crack propagation along the TD in the RD–TD plane leads to the crack-divided fracture type with relatively low fracture toughness.

It was recently shown [8,10] that the additional alloying of a high-strength low-alloy (HSLA) steel by carbonitride-forming elements such as V, Nb, and Ti increases strength and impact toughness after tempforming. A novel UHS steel with a chemical composition shown in Table 1 was recently developed [11]. This steel exhibited a YS of 1690 MPa, an ultimate tensile strength (UTS) of 2040 MPa, an elongation-to-failure of 8%, and a Charpy V-notch impact energy of 14 J after full austenitizing, quenching, and tempering at 280 °C. We expect that the tempforming may increase the strength, ductility, and toughness of this steel concurrently. The first aim of the present study is to evaluate the effect of tempforming by plate rolling on the mechanical properties, structure, and dispersion of the secondary phase in the novel UHS steel. After tempering at 600 °C (Q-T), this steel exhibited a YS of 1460 MPa, UTS of 1580 MPa, elongation-to-failure of 11.5%, and impact energy of 26 J. The second aim of the present study was to consider the microstructure and mechanical properties of this steel after it is subjected to tempforming and Q-T.

It is worth noting that the medium-carbon low-alloy steel considered as prototype for a novel UHS steel (Table 1) was subjected to tempforming at 500 °C [4], which requires a relatively high rolling force. An increase in tempforming temperature improves the technological efficiency of this TMP and enhances the workability of a UHS steel to avoid lateral cracking, which is important for the implementation of this technique into commercial use. The third aim of this paper is to evaluate the feasibility of increasing the tempforming temperature due to V, Nb, and Ti additions to the medium-carbon low-alloy steel (Table 1), improving retention strength and toughness.

## 2. Materials and Methods

A steel with a chemical composition as shown in Table 2 was subjected to hot forging at a temperature of 1150 °C (Figure 1a). Two types of billets were machined for the tempforming process. The first, with dimensions of 27 mm × 23 mm × 120 mm (thickness × width × length), was used for rolling up to the true strain of 0.5 and 0.89. The second, with dimensions of 45 mm × 32 mm × 153 mm, was used for rolling up to the true strain of 1.4. The billets were austenitized at a temperature of 900 °C for 40 min followed by water quenching. Next, the billets were tempered for 2 h at temperatures of 500, 550, 600, and 650 °C, followed by rolling at the same temperatures (Figure 1a) to total strains of 0.5, 0.89, and 1.4 (reduction in thickness of 39, 59, and 75%, respectively). A two-roller mill with a force capacity of 250 tons was used for warm rolling.

The Rockwell hardness tests were conducted using a Wolpert Wilson hardness tester 600 MRD (Illinois ToolWorks Inc., Norwood, MA, USA) in accordance with the ASTM E 18 standard [12]. At least five measurements of each specimen were conducted. The relationship between tensile test specimens and Charpy V-notch specimens and rolling direction is shown in Figure 1b. The RD-standard tensile tests were carried out at room temperature using flat specimens with a cross-section area of 7 × 3 mm^2^ and a 35 mm gauge length on an Instron 5882 testing machine (Illinois ToolWorks Inc., Norwood, MA, USA). At least three standard-tensile specimens of each condition were tested. In order to estimate tensile properties in three directions of tempformed billets, the mini-tensile specimens with a cross-section area of 1 × 0.6 mm^2^ and a 4 mm gauge length were used. RD-mini-tensile sample specimens were tested for comparison with RD-standard specimens and confirmation of the tensile tests results was obtained on mini-tensile specimens. At least five mini-tensile specimens of each condition were tested to estimate the scatter of the mechanical properties.

Strain measurements during all the tensile tests were carried out by the digital image correlation (DIC) method, which provides a high spatial resolution [13]. Before tests, one surface of the specimen was lightly coated with white spray paint and then oversprayed with a dark mist to obtain a black and white random pattern on the surface. The DIC method consists of monitoring the displacements of the speckles on the specimen surface during deformation, where the speckles are used as a mesh for the calculation of the strain. The DIC system provides the synchronization of the output signal from the load cell of the testing machine with the series of images. Such synchronization allows for an accurate plot of the tensile curve. The calculations were performed using Vic-2D software (Correlated Solutions, Inc., Irmo, SC, USA).

Standard Charpy V-notch (CVN) specimens were tested at room temperature using the Instron SI-1M impact machine with a maximum energy of 450 J with the Instron Dynatup Impulse data acquisition system (Instron corporation, Grove City, PA, USA) following the ASTM E 23 standard [14] in ND (crack-arrest type fracture) and TD (crack-divider type fracture) (Figure 1b). At least two specimens were tested after each processing route.

Structural characterizations were carried out using a JEM-2100 transmission electron microscope (TEM) (JEOL Ltd., Tokyo, Japan) and the Quanta 600 FEG scanning electron microscope (FEI Corporation, Hillsboro, OR, USA) incorporating an orientation imaging microscopy (OIM) system. TEM foils were prepared by double-jet electropolishing using a solution of 10% perchloric acid in glacial acetic acid. The transverse lath/subgrain sizes were measured from TEM micrographs by the linear intercept method, including all clearly visible sub-boundaries. The dislocation densities were evaluated by counting individual dislocations in the grain/subgrain interiors, and each data point represents at least six arbitrarily selected representative TEM images that were obtained using two-beam conditions with {002} or {013} scattering planes and a small positive deviation parameter.

The lamellar structure observations were conducted with EBSD (electron backscatter diffraction) and were focused on the transverse cross-section (RD × ND plane) of the rolled sheet (Figure 1b). EBSD was performed with a relatively fine scan-step size of 30 nm. To enhance the reliability of the EBSD data, the fine grains comprising three or fewer pixels were automatically removed from the maps using a standard grain-dilation option of the EBSD software. Furthermore, to eliminate the spurious boundaries associated with orientation noise, a lower-limit misorientation cut-off of 2° was applied. A 15° criterion was employed to differentiate low-angle boundaries (LABs) from high-angle boundaries (HABs).

Other details of the mechanical and structural characterization were described in previous papers [7,8,10,11].

## 3. Results

### 3.1. Rolling Force of Tempforming

At 500 °C, rolling results in a delamination of the billet along the center after a total rolling reduction of 59% and, therefore, tempforming at this temperature is not feasible. At 550 °C, lateral cracking appears after a rolling reduction of 59%. The rolling force of tempforming at different temperatures at a rolling reduction of ~8% for one pass is summarized in Table 3. It is seen that at temperatures ranging from 500 to 600 °C, the rolling force is relatively high. Increasing the rolling temperature to 650 °C results in a drop in rolling force that allows for increasing strain to be imposed on the steel for one pass. Therefore, temperatures of 600 and 650 °C are feasible for tempforming and the use of 650 °C allows for a significant decrease in rolling force.

### 3.2. Microstructure after Tempering

The microstructure after quenching and tempering at 600 °C was described in a previous paper [11] in detail. The tempered martensite lath structure (TMLS) with a size of prior austenite grains of ~12 μm, a distance between high-angle boundaries (HABs) of 0.84 μm, a lattice dislocation density of ~6 × 10^14^ m^−2^, and a lath thickness of ~200 nm evolved after tempering at 600° [11]. Tempering at T > 500 °C produces M_6_C and M_23_C_6_ carbides located on HABs and low-angle boundaries (LABs) and a dispersion of MX carbonitrides located in a ferritic matrix [11]. M_6_C, M_23_C_6_, and MX particles exhibit a round shape and dimensions of ~210, ~50, and ~45 nm, respectively [11].

### 3.3. Microstructure after Tempforming

Electron backscatter diffraction (EBSD) maps with {100} pole figures, TEM micrographs, and the orientation distribution function (ODF) of microstructures after different tempforming routes are shown in Figure 2, Figure 3 and Figure 4. Warm rolling after tempering at the same temperatures leads to the transformation of TMLS to a lamellar structure with increasing rolling reduction (Figure 2 and Figure 3). The reorientation of HABs and LABs to make them parallel with RD and the elongation of packets, blocks, and laths along RD takes place. At tempforming temperatures of 600 and 650 °C, the distinct lamellar structure evolves after total strains of ~0.89 and ~0.5, respectively (Figure 2a,b).

The lattice dislocation density increases by 40% in comparison with quenched and tempered steel [11] (Table 4). Spacing between LABs, which roughly corresponds to transverse lath size, decreases by a factor of ~2 compared with the thickness of the initial laths in quenched and tempered steel [11] (Table 4, Figure 3). At tempforming temperatures of 600 and 650 °C, the spacing between HABs decreases by a factor of more than 2 and 1.3, respectively. Packets with an irregular shape retain their structure and the inclination angle ranges from 10 to 60°. This finding is in contrast with low-carbon steel [15].

After tempforming at temperatures of 600 and 650 °C with total strain of ~1.4, the lamellar structure evolves (Figure 2c,d). The lattice dislocation density increases, although not significantly, attaining a high value of 10^15^ m^−2^ (Table 4). The angle of inclination to RD is <10° and the lamellar boundaries are straight. The TEM observations showed that boundary carbides effectively pin HABs and LABs, suppressing their migration (Figure 3). It should be noted that low-angle boundaries with misorientations of more than 2 degrees are shown in the EBSD map (Figure 2). However, laths could not be distinguished in the maps. Therefore, most of the boundaries on TEM photos in Figure 3 have misorientations of less than 2 degrees. An extensive transformation of extended LABs to HABs occurs under tempforming at 650 °C, which leads to close spacings between HABs and LABs, and an increasing proportion of HABs with increasing rolling reduction (Table 4). It is evident that a type of continuous dynamic recrystallization (cDRX) provides the transformation of LABs to HABs during warm rolling [16]. Boundary M_6_C and M_23_C_6_ carbides and MX carbonitrides located in the ferritic matrix effectively pin the lamellar boundaries, providing the necessary conditions for the transformation of LABs to HABs under warm rolling [16].

At a tempforming temperature of 600 °C, cDRX occurs slowly and the average spacing between HABs is higher than the average spacing between LABs by a factor of ~4 (Table 4). A relatively high proportion of LABs is retained in the lamellar structure and the average misorientation increases, although not significantly, with increasing rolling reduction (Table 4). Lattice dislocation density slightly increases with increasing rolling reduction (Table 4) and, therefore, the occurrence of cDRX is accompanied by an increase in lattice dislocation even at a tempforming temperature of 650 °C. This is a specific feature of the cDRX occurrence under tempforming. Lattice dislocation densities after tempforming and water quenching are essentially the same in the steel studied here [11]. Therefore, tempforming strongly reduces the spacing between LABs/HABS, reorients these boundaries along RD, and increases lattice dislocation density up to a value observed after water quenching.

In order to examine the crystallographic texture that evolved after tempforming, orientation data were extracted from EBSD maps in Figure 2 and arranged as ODFs. The characteristic φ_2_ = 45° sections of the ODFs are summarized in Figure 4. In both material conditions produced after the true strains of 0.89 and 0.5, the developed textural patterns were inconsistent with the typical rolling textures in bcc metals (Figure 4a,b). This observation suggests that the process of texture formation was not completed, presumably due to the insufficiently low rolling strain. Hence, the evolved textures were likely represented by transient orientations from the initial transformation texture to the final rolling texture. After rolling to a true strain of 1.4, the texture was dominated by a superposition of the <110>//RD and <111>//ND components, i.e., the alpha and gamma fibers (Figure 4c,d). These textures are typically observed in heavily rolled bcc metals.

In the material rolled at 600 °C, the gamma fiber (with the pronounced $\left(111)[0\bar{1}1\right]$ component) was found to predominate (Figure 4c). In contrast, rolling at 650 °C promoted a preferential formation of the alpha fiber with a prominent $\left(001)[1\bar{1}0\right]$ rotated cube orientation (Figure 4d). This textural component is often observed in plate-rolled bcc metals and alloys [17,18], as well as after tempforming [4,7,8,9].

The analysis of secondary phase particles (not shown here) showed no effect of tempforming on a dispersion of carbides and carbonitrides. Boundary M_6_C and M_23_C_6_ carbides comprise chains along lamellar boundaries and MX carbonitrides are located in the ferritic matrix (Figure 3). The dimensions of three types of particles after tempering [11] and subsequent warm rolling are the same (not shown here).

### 3.4. Tensile Tests

Engineering stress–strain curves obtained by tension in the RD, ND, and TD are presented in Figure 5. The average values with their standard deviations of yield stress (YS), *S_y_*, ultimate tensile strength (UTS), *S_u_*, total elongation, *El_t_*, and Rockwell hardness are summarized in Table 5. In order to estimate the properties of the sheet after tempforming in three directions, the mini-tensile test specimens were used. The results of statistical analysis are collected in Appendix B.

Large scattering in YS and UTS is observed in the steel tempformed at a temperature of 600 °C with a total strain of 1.4 both in the RD and TD (Table 5, Figure 5a,b). The tensile curves obtained on the RD–standard-tension test specimens were located under the curves obtained on the RD–mini-tensile test specimens. The minimum of UTS was measured to be 1790 MPa and exceeded the value UTS of 1580 MPa, which was obtained after Q-T treatment with a tempering temperature of 600 °C [11]. Large scattering of the UTS probably results from high internal stresses and heterogeneous microstructure after tempforming at 600 °C. Total elongation in the TD was more than two times less than that in the RD (Table 5). The ND mini-tensile test specimens display a completely brittle fracture, which occurs in the elastic region without any plastic strain (Figure 5c). The values of UTS in the range from 630 to 1190 MPa with the median of 750 MPa were obtained from the tests of ND mini-tensile specimens (Appendix B). It should be pointed out that the fracture stresses obtained by dividing the load at fracture by the minimum neck area of the ruptured specimen may be of little significance, because this method of calculation does not take into account a stress concentration in the neck. The crack in an unnotched tensile specimen is usually initiated at its axis and propagates toward the surface. The neck is formed when the metal at the periphery extends longitudinally and contracts radially during fracture. However, the brittle specimens contain no neck due to the absence of plastic strain; therefore, the UTS of specimens fractured in elastic regions could be considered to be close to the fracture stresses which lead to cleavage crack formation in RD.

Figure 5c,d shows that tensile curves obtained on the RD–standard-tension test specimens and RD–mini-tensile-test specimens after tempforming at 650 °C coincide with each other. The scatter of the data is much lesser in comparison with steel after tempforming at 600 °C with a total strain of 1.4. The UTS remarkably decreases and the ductility noticeably increases in all directions when the tempforming temperature increases from 600 to 650 °C (Table 5).

The shape of engineering stress–strain curves observed after rolling at 600 °C is typical for UHS steel with an ultra-fine-grained structure and could be classified as Type II in accordance with the classification of Yu et al. [19,20]. Type II curves exhibit a discontinuous yielding type, for which in the beginning a drop in yield occurs after the maximum load point and then the load decreases until rupture without work-hardening (Figure 5a,b).

The shape of the curves after tempforming at 650 °C exhibits a Type III behavior, for which in the beginning a drop in yield appears, followed by the Lüders deformation under constant load; then, work-hardening continues up to the maximum load point, and the load decreases until rupture (Figure 5d–f). It should be noted that strain hardening is small and the UTS is slightly higher than the YS after tempforming at 650 °C. Thus, the tempforming at a temperature of 650 °C provides the steel with ductility in all directions, while after tempforming at a temperature of 600 °C the steel becomes brittle in an anisotropic manner.

Values of hardness after tempforming (Table 5) are slightly higher than after tempering at the corresponding temperatures [11]. The increase in hardness correlates with an increase in UTS after tempforming.

### 3.5. Impact Toughness

The load–deflection curves are shown in Figure 6, and typical CVN specimens after impact tests with corresponding values of absorbed impact energy are presented in Figure 7 for the ND and the TD. The average values of absorbed impact energies with standard deviations are collected in Table 6. The results of statistical analysis of impact properties are collected in Appendix C. The CVN absorbed energy and the shape of the load–deflection curves of TD-Charpy V-Notch specimens remain the same as after ordinary quenching and tempering treatment [11]. The similar values of impact energy are obtained both at room temperature and at the low temperature of −40 °C. Therefore, tempforming has a minimal effect on the impact properties along the TD.

A two-fold increase in impact toughness is observed along the ND after warm rolling at temperatures of 600 and 650 °C, with reductions in thickness of 59% (ε = 0.89) and 39% (ε = 0.5), respectively (Figure 7, Table 6). The shape of the load–deflection curves obtained along the ND is the same as those along the TD, while two peaks could be observed due to the strong delamination of the specimen (Figure 6c). No general yield load, P_GY_ [21], could be distinguished after rolling reductions of 59% (ε = 0.89) and 39% (ε = 0.5).

The increase in rolling reduction up to 75% (ε = 1.4) led to an increase of one order of magnitude in room temperature impact toughness along the ND, in comparison with the TD (Figure 7). Moreover, the low temperature tests at −40 °C display the same extremely high values of impact toughness. The load–deflection curves obtained in the ND and the shape of the fractured CVN specimens are typical for delamination toughness [4,5,6,7,8,9,10]. Irrespective of the test temperature, the ND-Charpy V-Notch specimens exhibit four stages of fracture. The general yield of the specimen starts in the first stage of fracture at a general yield load (P_GY_). In the second stage, the main crack initiates between P_GY_ and the maximum load (P_max_) [21]. The third stage is stable crack propagation. The inspection of fractured specimens in Figure 7 shows that all ND-Charpy V-Notch specimens contain secondary cracks near the notch in the transverse direction of the main crack. Those cracks appear after a short stage of the stable propagation of the main crack and lead to the disappearance of the stress concentration at the tip of the main crack. Next, at the fourth stage of fracture, the lamellar structure is bent as an unnotched specimen until the plastic deformation of the outer layer of the lamellar structure locally reaches elongation-to-failure and a new macrocrack is nucleated along the ND. The load sharply decreases at P_F_ when a new macrocrack starts to propagate. This fracture process can repeat and several secondary cracks are formed in the specimen. Thus, high CVN energy after warm rolling is attained in ND–Charpy V-Notch specimens due to the appearance of prolonged stages of bending the specimen with a secondary crack, which serves as barrier to the propagation of the main crack.

### 3.6. Fractography

Figure 8 and Figure 9 show the fractographs of the CVN specimens after impact tests in the TD and ND, respectively. Observations of fractured surfaces in the TD show a sawtooth fracture, which is a result of the delamination process (Figure 8a). Long parallel deep splits propagate along the rolling plane, i.e., in accordance with the impact direction (Figure 8a,b). Numerous secondary cracks and small dimples are observed between the splits (Figure 8b,c). The surfaces of splits and secondary cracks represent cleavage facets (Figure 8c). Thus, the delamination is initiated on parallel cleavage {100} planes lying almost parallel to the RD.

The fractures surfaces of the ND–Charpy V-Notch specimens are rather different from those of the TD specimens (Figure 9). The main cracks propagate directly across the central portions of the impact test bars until delamination cleavage initiates in the perpendicular direction to the displacement of the pendulum parallel to the RD–TD plane (Figure 9a,d). The length of the zone of the main crack initiation and propagation before delamination is short (Figure 9a,d).

Terraces are formed on the delamination fracture surface roughly parallel to the RD and steps are formed on the surfaces roughly parallel to the ND (Figure 9b). Terraces represent transgranular cleavage facets, while steps are surfaces with fine ductile dimple patterns (Figure 9c). It should be noted that stepwise crack propagation was revealed in other steels, exhibiting similar delamination toughening behavior [22,23]. After delamination, the propagation of the main crack is arrested because the flat terrace eliminates the radius curvature at the tip of the main crack and the stress concentration decreases. Then, plastic deformation by bending effectively consumes impact energy until new main crack nucleates due to the high strain level and the process of delamination fracture repeats. The main crack propagates by the ductile mechanism, because cross-sections of broken lamellas consist of fine ductile dimple patterns (Figure 9e). The fracture mechanisms of specimens after rolling with ε = 0.89 and ε = 1.4 are similar. However, the fraction of the (100) plane along the RD increases with increasing rolling reduction (Table 4); therefore, the probability of the formation of terraces in the crosswise direction to the main crack propagation is higher in specimens with ε = 1.4. As a result, main crack arrest by terrace formation is more effective in specimens after tempforming with ε = 1.4.

## 4. Discussion

The experimental results show that the steel presented here meets the requirements for UHS steels after tempforming at a temperature of 650 °C with a total strain of 1.4. Tempforming in these conditions increases strength and toughness concurrently, while ductility remains virtually unchanged. Tempforming at 600 °C provides high strength but the steel is brittle in one direction and exhibits low ductility. In addition, at 600 °C, the rolling force is relatively high. Carbide-forming elements provide a ~100 °C increase in the tempforming temperature of the present steel in comparison with the medium-carbon low-alloy steel (Table 1) without strength loss [4]. V, Nb, and Ti additives increase the lowest temperature feasible for warm rolling by ~100 °C. The steel exhibits sufficient workability at a rolling temperature of 650 °C.

### 4.1. First Type of Delamination

Two types of delamination toughness corresponding to ND–Charpy V-Notch specimens and RD–Charpy V-Notch specimens were found in the rolled plate (Figure 1b). The first type of delamination provides very high toughness in the ND, while the second type of delamination affects the absorbed energy of CVN impact, although not significantly. This difference is caused by the effect of microstructure and texture on the nucleation and propagation of secondary cracks. Fractography observations show that the propagation of the primary crack occurs by ductile dimple fracture mechanisms, while the propagation of secondary cracks takes place through a quasi-cleavage mechanism. Therefore, there is a strong difference in brittle fracture stress, σ_F_ [4,24,25], in mutually perpendicular directions. Brittle fracture stresses are dependent on effective grain size, d_eff_, and the surface energy of the {100}_α_ cleavage plane [4,24].

It is known that the effect of effective grain size on the fracture stress for brittle/quasi-brittle mode can be expressed as follows:(1)$$\sigma_{F}=1.41\sqrt{\frac{2{E\gamma }_{s}}{\pi \left(1-\nu^{2}\right)}}d_{\mathrm{eff}}^{-0.5}$$
where E = 212 GPa is Young’s modulus, γ_s_ = 34 J/m^2^ is the surface energy of the cleavage plane for the ferrite/carbide structure as in the present steel, and ν = 0.293 is Poisson’s ratio [4,24,25]. We can take the spacing between HABs as d_eff_ for the evaluation of the ratio between fracture stress in the RD and ND. The average spacing between HABs along the ND is shown in Table 4 and comprises between 0.28 and 0.195 μm after tempforming with ε = 1.4 at 600 and 650 °C, respectively. The average spacing between HABs along the RD is difficult to accurately measure, but could be estimated from EBSD maps in Figure 2 as 6 μm for specimens after tempforming with ε = 1.4 at 600 and 650 °C. Fracture stress along the RD was calculated to be 1.3 GPa for the steel after tempforming at temperatures of 600 and 650 °C. Rather higher fracture stresses of 5.9 and 7.0 GPa were calculated in the ND for tempforming temperatures of 600 and 650 °C, respectively. The σ_F_ values in the ND are higher by a factor of ~5 than those in the RD (Figure 10a). It is worth noting that we took the same γ_s_ value for two tempforming temperatures for the sake of simplicity. However, it is known [24,25] that the σ value is strongly dependent on the carbon content in low-alloy steel, and the depletion of carbon from α-solution with an increasing temperature of tempering has to increase the surface energy, and, therefore, brittle fracture stress.

The dynamic yield strength, σ_dY_, can be derived from the P_GY_ value using the following relationship assuming the Tresca yield criterion [26]:(2)$$\sigma_{\mathrm{dY}}=\frac{2.99P_{\mathrm{GY}}W}{B{\left(W-a\right)}^{2}}$$
where W and B are the test bar width and thickness, respectively (10 mm), and a is the notch depth (2 mm). The σ_dY_ values were calculated as 1.45 and 1.84 GPa for tempforming temperatures of 600 and 650 °C, respectively. The criterion for the onset of cleavage crack propagation is expressed as [26]
(3)$$\sigma \geq \sigma_{F}$$
where σ is the stress normal to direction of crack propagation related to σ_dY_ through the intensification factor, Q, as σ_yy_ = Qσ_dY_. We take Q = 1 for the sake of simplicity. Therefore, fracture stress along the RD is significantly higher than the σ_dY_ values and the propagation of the primary crack occurs in a ductile manner, while along the ND the values of fracture stresses are smaller than the dynamic yield stress. The criterion described by Equation (3) is fulfilled and secondary cracks propagate by quasi-cleavage mechanism.

The first type of delamination is observed for the propagation of primary cracks in the ND. This type of fracture provides very high delamination toughness in this direction and is characterized by the following sequence of crack propagation stages (Figure 10b,c):The propagation of the primary crack occurs in the ND in a stable manner by ductile fracture (Figure 10b);The nucleation and propagation of secondary cracks by the cleavage mechanism in the RD along lamellas with a {100}_α_ <110> orientation occur at the stage of stable crack propagation;The main crack tip is blunted due to secondary crack propagation and the decrease in stress concentration in the vicinity of the main crack (Figure 10c);The plastic deformation of the lamellar structure takes place until a new main crack forms due to fracture stress in the outer layer of the bending specimen;Load drop occurs due to the nucleation and propagation of a new main crack in bended lamellas.

The main feature of this delamination toughness is the nucleation of secondary cracks at the tip of the primary crack. As a result, the tip of the primary crack becomes blunt, a triaxial tension state in the vicinity of the crack tip transforms to a uniaxial one, and crack propagation in the ND is suppressed by delamination. The resumption of crack propagation along the ND is possible after the plastic deformation of the outer layer of the bending specimen and the nucleation of the new main crack. Plastic deformation effectively dissipates the impact energy and increases the impact toughness. The probability of the formation of new cleavage delamination cracks depends on the number of grains with {100}_α_ <110> orientation. Increasing rolling reduction increases the number of grains with texture {100}_α_ <110>. Therefore, impact toughness increases in a highly deformed sheet due to main crack arrest by cleavage delamination cracks.

The second type of delamination is observed in the TD. The nucleation of first-order secondary cracks occurs on some points of the primary crack and the tip of this crack remains sharp, providing a high rate of propagation. This type of delamination has no effect on CVN absorbed energy since this delamination is a process that accompanies primary crack propagation. The effect of secondary crack nucleation on the propagation of the primary crack is insignificant. Thus, the delamination toughness is attributed to the strong effect of nucleation and the propagation of secondary cracks on the propagation of the primary crack.

### 4.2. Second Type of Delamination

The effect of rolling reduction and tempforming temperature on delamination is attributed to its effect on the coherent length of cleavage fracture equal to the dimension of continuous bands consisting of lamellas with {100}_α_ <110> orientation along the RD [3,4,5,6,7,8,9,10]. Cleavage easily occurs along {100}_α_ planes crossing HABs with chains of boundary M_23_C_6_ and M_6_C carbides [27] without deflections at a distance of the coherent length up to attaining a lamella with another orientation playing the role of an obstacle to quasi-brittle fracture. Therefore, a distance of cleavage along {100}_α_ in adjacent lamellas with {100}_α_ <110> orientation along the RD determines the length of the cleavage terraces. It seems that changes in the crystal lattice orientation of adjacent lamellas are more important for propagating cleavage fracture than such imperfections as HABs, LABs chains of boundary carbides, and matrix carbonitrides [28]. A lamella with an orientation distinctly distinguished from {100}_α_ <110> orientation along the RD plays a role of insurmountable obstacle for cleavage propagation. If the transverse size of a lamella is small, the cleavage may bypass a lamella with another orientation by fracture propagation in a ductile manner, up to the nearest lamella with a {100}_α_ <110> orientation along the RD. The thickness of lamellas with another orientation determines the height of the steps. The alternation of lamellas with {100}_α_ <110> orientation along the RD with lamellas with another orientation determines the stepwise shape of secondary cracks and the length of secondary crack propagation. Therefore, the longitudinal dimensions of lamellas with {100}_α_ <110> orientation along the RD and the proportion of lamellas with this orientation among all lamellas control the coherent length of cleavage fracture. It is worth noting that the elongation of lamellas along the RD is important to attaining low values of fracture stress in accordance with Equation (1). No cleavage may occur in short lamellas with {100}_α_ <110> orientation along the RD since the σ_F_ value may exceed the dynamic yield strength.

Thus, the lamella aspect ratio (AR), defined as the ratio of the dimensions between HABs in the RD to that in the TD/ND, with a {001}_α_ <110> (rotated cube) texture component is extremely important for delamination toughness. At low rolling reductions, AR and the number of lamellas with {100}_α_ <110> orientation along the RD are relatively low and delamination belonging to the first type occurs rarely due to the small coherent length of cleavage fracture. In addition, lamellas located at an inclination angle to the RD and the mixed transgranular fracture occur as they do under the tempered condition [11]. Fracture stress may be smaller than dynamic yield strength and quasi-brittle fracture takes place under the propagation of the primary crack in local areas. Therefore, the formation of strong structural and crystallographic textures is a necessary condition to suppress brittle fracture for the propagation of a primary crack. High σ_dY_ values could be attained due to the fully ductile fractures of the primary crack path, which allow for delamination with a high coherent length of cleavage fracture in a perpendicular RD. The combination of these two factors provides a high CVN absorbed energy that is feasible after a high rolling reduction.

## 5. Conclusions

Tempforming at temperatures of 600 and 650 °C by plate rolling provides the formation of a lamellar structure in 0.4%C-2%Si-1%Cr-1%Mo-VNb steel with a spacing between high-angle boundaries of ~200 nm, lamella length of ~6 μm, and lattice dislocation density of 10^15^ m^−2^. M_6_C and M_23_C_6_ carbides with dimensions of ~200 and ~50 nm, respectively, are located on high- and low-angle boundaries and MX carbonitrides with a dimension of ~46 nm are located in the ferritic matrix.Warm rolling after tempering at the same temperatures provides the formation of a strong {001}<110> (rotated cube) component, the elongation of lamellas along the rolling direction, an increase in lattice dislocation, and the transformation of low-angle boundaries to high-angle ones.Tempforming at temperatures of 600 and 650 °C with a total strain of 1.4 provides upper yield stresses of 1775 and 1550 MPa, ultimate tensile strengths of 1810 and 1555 MPa, elongations-to-failure of 4.7 and 9.4%, and a Charpy V-notch impact absorbed energy in the ND of ~219 and ~242 J·cm^−2^, respectively.Two types of delamination were found for the propagation of the primary crack in the normal and transverse directions. Delamination toughness is associated with the nucleation of secondary cracks along the whole length of the primary crack and their propagation over large distances leading to the appearance of stepwise terraces. The critical condition for delamination toughness is the propagation of the primary crack by the ductile fracture mechanism due to very high fracture stress and the propagation of secondary cracks by the brittle quasi-cleavage mechanism due to relatively low fracture stress. A 6–7-fold difference in the values of fracture stress in the normal/transverse and rolling directions is attributed to the difference in spacing between high-angle boundaries in lamellas. Delamination with the nucleation of secondary cracks in several points along the overall length of the primary crack has little effect on the Charpy V-notch impact absorbed energy.

## Figures and Tables

**Figure 1 materials-16-01202-f001:**
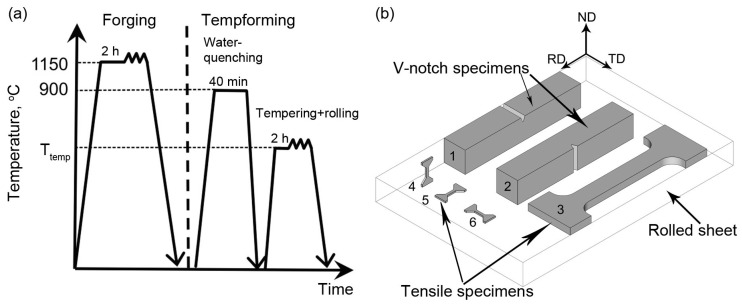
Schematic illustrations of the processing routes: (**a**) diagrams showing thermo-mechanical processing; and (**b**) the orientations for tensile and impact tests. 1—ND-Charpy V-Notch specimen, 2—TD-Charpy V-Notch specimen, 3—RD-standard-tension test specimen, 4—ND-mini-tensile test specimen, 5—RD-mini-tensile specimen, 6—TD-mini-tensile specimen.

**Figure 2 materials-16-01202-f002:**
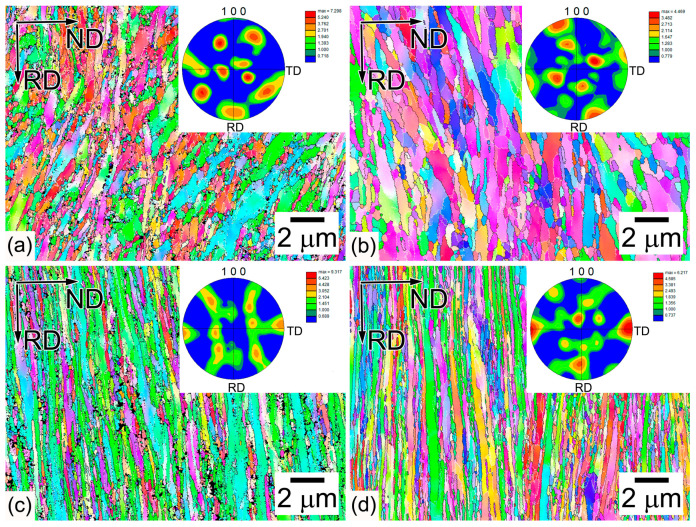
Microstructures in steel after tempforming at (**a**,**c**) 600 °C to a total strain of (**a**) 0.89 and (**c**) 1.4, and (**b**,**d**) 650 °C to a total strain of (**b**) 0.5 and (**d**) 1.4. The white and black lines mean boundaries with misorientations from 2 to 15 deg and above 15 deg, respectively.

**Figure 3 materials-16-01202-f003:**
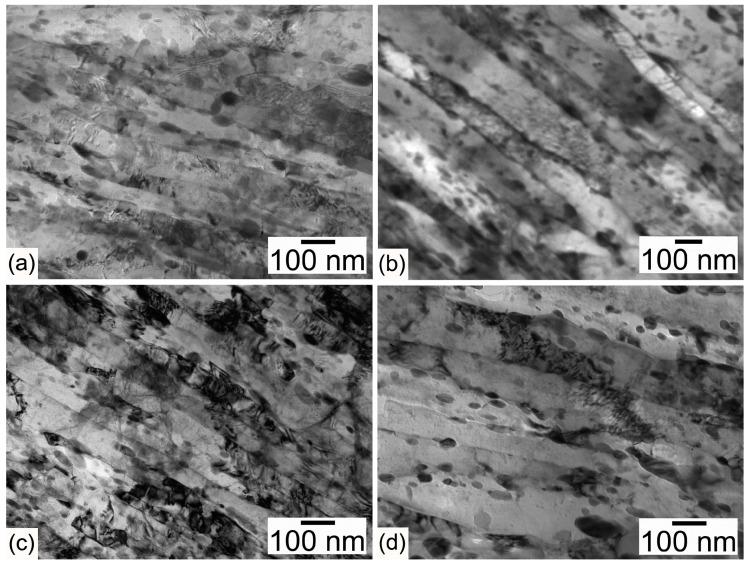
Typical TEM photographs of substructure in steel after tempforming at (**a**,**c**) 600 °C to total strains of (**a**) 0.89 and (**c**) 1.4 and at (**b**,**d**) 650 °C to total strains of (**b**) 0.5 and (**d**) 1.4.

**Figure 4 materials-16-01202-f004:**
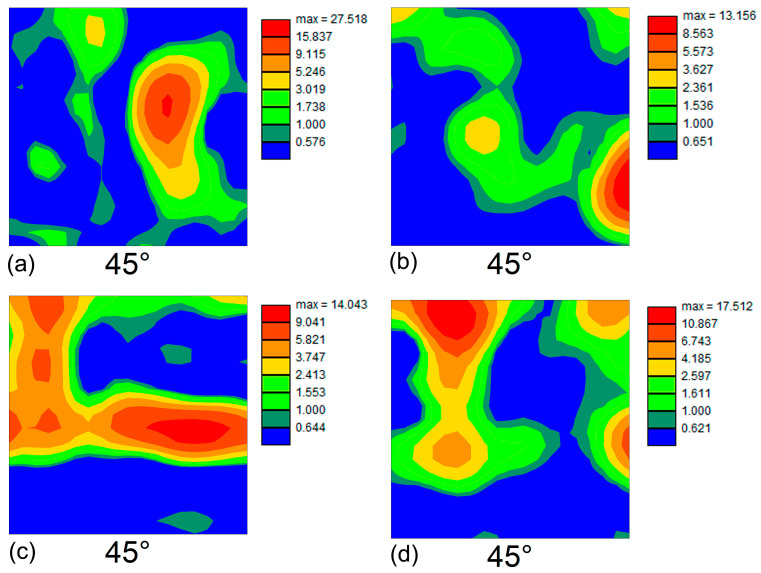
Orientation distribution functions (ODF) at φ_2_ = 0° of the steel after tempforming at (**a**,**c**) 600 °C to total strains of (**a**) 0.89 and (**c**) 1.4 and (**b**,**d**) at 650 °C to total strains of (**b**) 0.5 and (**d**) 1.4.

**Figure 5 materials-16-01202-f005:**
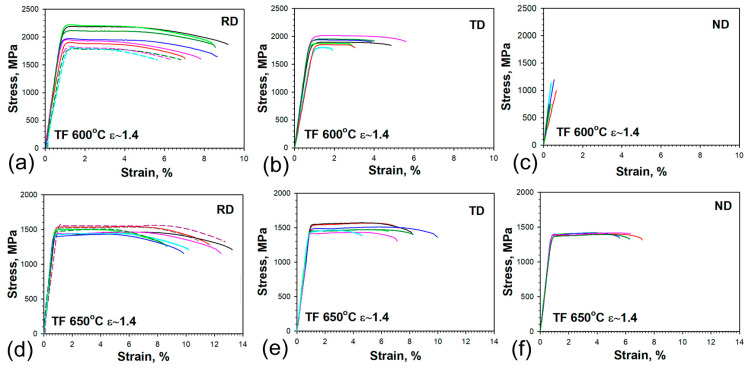
Engineering stress–strain curves of tempformed (TF) steel obtained for different orientations of tensile specimens at (**a**–**c**) 600 °C and (**d**–**f**) 650 °C. The continuous and dashed lines indicate engineering stress–strain curves obtained on the mini-tensile and standard specimens, respectively.

**Figure 6 materials-16-01202-f006:**
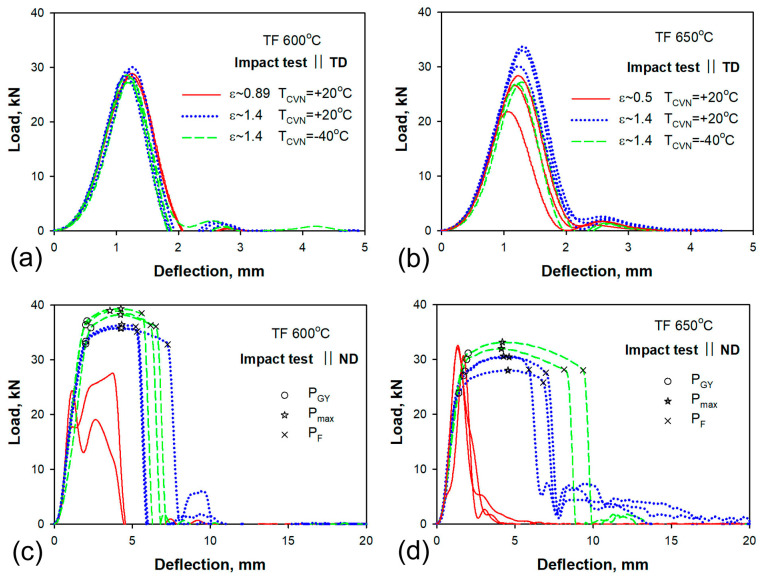
Load–deflection curves of tempformed (TF) steel: (**a**,**b**) TD-Charpy V-Notch specimen, (**c**,**d**) ND-Charpy V-Notch specimen. The line types for different rolling reductions (ε) and different test temperatures are indicated in inserts (**a**) for 600 °C and (**b**) for 650 °C.

**Figure 7 materials-16-01202-f007:**
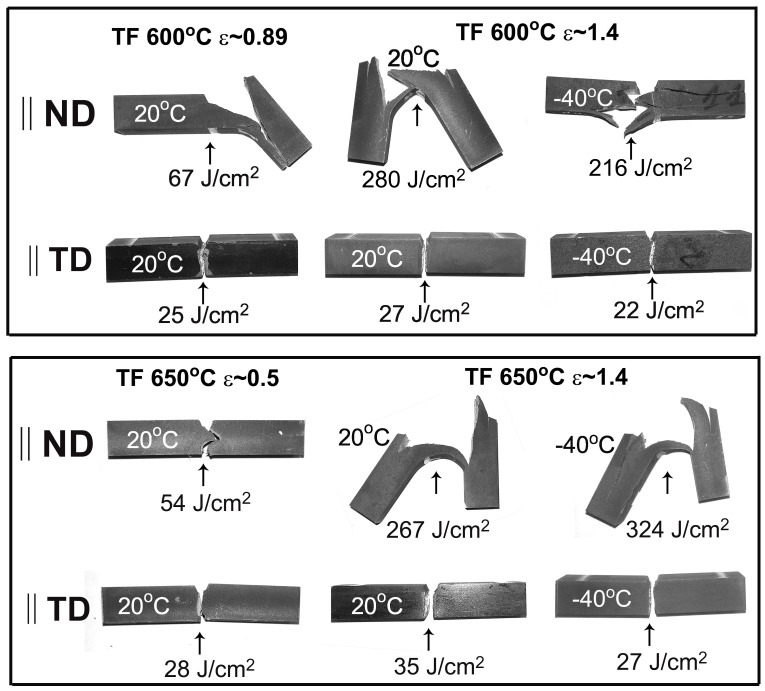
Typical Charpy V-notched specimens after impact tests and their impact absorbed energies. The steel was subjected to tempforming (TF) with different true strains.

**Figure 8 materials-16-01202-f008:**
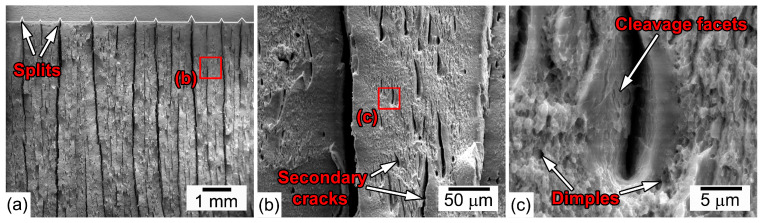
SEM images of typical sawtooth fracture surface of the steel after the impact test of the TD–Charpy V-Notch specimen at room temperature. Tempforming at 600 °C, ε = 1.4. (**a**) General view of fractured impact specimen, (**b**) secondary cracks between splits (**c**) high-magnification of typical secondary cracks morphology and surface around them.

**Figure 9 materials-16-01202-f009:**
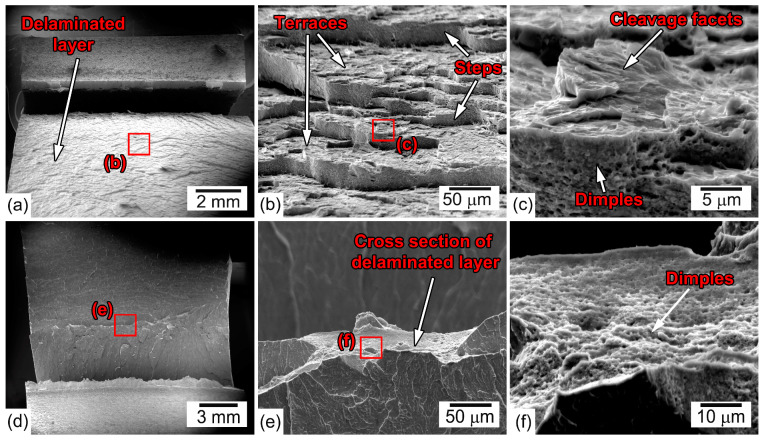
SEM images of fracture surface of the steel after the impact test of the ND–Charpy V-Notch specimen at room temperature. (**a**–**c**) tempforming at 600 °C, ε = 0.89, (**d**–**f**) tempforming at 600 °C, ε = 1.4.

**Figure 10 materials-16-01202-f010:**
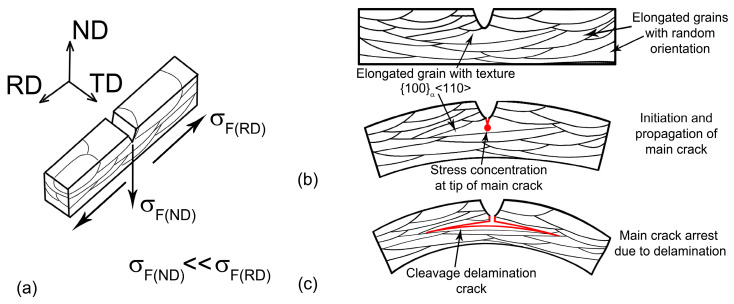
Schematic illustration of delamination in ND–Charpy V-Notch specimen after tempforming: (**a**) the effect of grain size on the fracture stress, (**b**) the nucleation and propagation of main crack at first stage of fracture, (**c**) formation of secondary crack in RD along lamellas with {100}_α_ <110> orientation.

**Table 1 materials-16-01202-t001:** Chemical composition of the medium-carbon low-alloy steel and novel UHS steel (weight percentage).

	C	Si	Mn	Cr	Mo	V	Nb	Ti	B	S	P	Fe
Medium-carbon low-alloy steel [4]	0.40	2.0	0.01	1.0	1.0	-	-	-	-	-	-	Bal.
Novel UHS steel [11]	0.43	1.60	0.01	1.10	0.95	0.08	0.05	0.04	0.003	0.007	0.004	Bal.

**Table 2 materials-16-01202-t002:** Chemical composition of the experimental steel (weight percentage).

C	Si	Mn	Cr	Mo	V	Nb	Ti	B	S	P	Fe
0.43	1.60	0.01	1.10	0.95	0.08	0.05	0.04	0.003	0.007	0.004	Bal.

**Table 3 materials-16-01202-t003:** Effect of tempforming temperatures on rolling force in tons at a rolling reduction of ~8% for one pass.

Tempforming Temperature, °C	500	550	600	650
Rolling force, tons ^1^	97	94	93	67

^1^ The initial billet dimensions of 27 mm × 23 mm × 120 mm (thickness × width × length).

**Table 4 materials-16-01202-t004:** Microstructure parameters of steel processed by tempforming.

Tempforming Route	Distance between HAB in ND, µm	Fraction of HABs, %	Average Boundary Misorientation	Fraction of (100) along RD	Transverse Lath Size, nm	Dislocation Density, 10^14^ m^−2^
600 °C ε~0.89	0.385	52	26	10.8	106 ± 53	9.3 ± 2.7
600 °C ε~1.4	0.280	60	27	19.9	72 ± 29	10.0 ± 3.8
650 °C ε~0.5	0.623	66	34	12.2	117 ± 67	8.6 ± 2.7
650 °C ε~1.4	0.195	72	34	36.5	127 ± 65	11.0 ± 6.5

**Table 5 materials-16-01202-t005:** Average values of the static mechanical properties and their standard deviations of the tempformed steel along different directions.

Tempforming Route	Specimen Type ^1^	*S_y_*, MPa	*S_u_*, MPa	*El_t_*, %	*HRC*
600 °C ε~1.4	RD–standard-tension	1775 ± 30	1810 ± 20	4.7 ± 0.4	49.5 ± 1.5
	RD–mini-tensile	2020 ± 130	2060 ± 135	8.5 ± 1.1	-
	TD–mini-tensile	1840 ± 75	1905 ± 75	3.5 ± 1.4	-
	ND–mini-tensile	-	-	-	-
650 °C ε~1.4	RD–standard-tension	1550 ± 30	1555 ± 35	9.4 ± 1.0	45.5 ± 1.5
	RD–mini-tensile	1435 ± 55	1475 ± 40	11.5 ± 1.8	-
	TD–mini-tensile	1450 ± 45	1505 ± 60	7.6 ± 1.8	-
	ND–mini-tensile	1375 ± 25	1410 ± 10	5.3 ± 1.4	-

^1^ Designations of specimens are in accordance with the schematic illustration in Figure 1b.

**Table 6 materials-16-01202-t006:** Average absorbed impact energies and their standard deviations of Charpy V-Notch specimens in different directions of the tempformed steel.

Tempforming Route	Specimen Type ^1^	CVN, J/cm^2^	
		+20 °C	−40 °C
600 °C ε~0.89	ND	77 ± 14	-
	TD	25 ± 1	-
600 °C ε~1.4	ND	219 ± 53	227 ± 19
	TD	26 ± 1	22 ± 1
650 °C ε~0.5	ND	48 ± 8	-
	TD	27 ± 5	-
650 °C ε~1.4	ND	242 ± 22	300 ± 35
	TD	35 ± 2	26 ± 1

^1^ Designations of specimens are in accordance with schematic illustration in Figure 1b.

## Data Availability

Not applicable.

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
