# Peer review of "Effect of Tempforming on Strength and Toughness of Medium-Carbon Low-Alloy Steel"

_materials, 2023, doi:10.3390/ma16031202_

Round 1
Reviewer 1 Report
The present paper studies the microstructure and mechanical properties during the tempforming of a medium carbon low alloy steel, namely, austenization+quenching+tempering+plate rolling at tempering temperature. It is reported that plate rolling followed by tempering at the same temperature of 600oC increases the yield strength by 25%, the Charpy V-norch impact energy by a factor of ~10. The underlying mechanism responsible for the mechanical properties in particular the impact toughness was proposed. The present paper present interesting results and can be considered for publication in case the following comments were properly replied.
1) Experimental details by EBSD examination should be given.
2) The dislocation density measured by TEM method should detailed.
3) The separation of low angle boundaries from high angle boundaries on the TEM image should be presented.
4) Figure 4 was presented but not mentioned in the text. From Fig. 4, the boundary spacing for (d) 650oC 1.4 seems bigger than that of the rest (a), (b) and (d), whereas the Table 2 gives opposite results.
5) Low temperature toughness test was made and shown in Figures 6 and 7, but was not discussed in this paper.
6) For equation (1), the effective grain size should be clarified in more detail. Why the effective grain size was used 10 micron for RD but 0.28 and 0.2 micron for ND/TD?
7) 3.6 Fractography should be rewritten. The fracture process should be based on the evidences given by the structural characterization rather than the literatures. The so many nomenclatures like delamination, delamination terraces, first-order terraces, crack arrester, etc should have direct experimental evidences.
8) Language and expression should be enhanced so as to enhance the readability.
Author Response
- Experimental details by EBSD examination should be given.
Experimental details of EBSD examination have been added in revised manuscript.
- The dislocation density measured by TEM method should detailed.
Experimental details of dislocation density measurement have been added in revised manuscript.
- The separation of low angle boundaries from high angle boundaries on the TEM image should be presented.
Unfortunately, the study of grain misorientations by Kikuchi lines technique on TEM is out scope of this paper. We revised the description of measurement details of lamellar structure. The transverse lath/subgrain sizes were measured instead of distances between all clearly visible low-angle boundaries. It should be noted that low angle boundaries with misorientation more than 2 degree measured by EBSD analysis and maps is shown in Figure 2. However, small laths with size of 120 nm, which were revealed by TEM analysis, are not visible in those maps. Therefore, most of boundaries in TEM photos in Figure 2 have misorientations less than 2 degree.
- Figure 4 was presented but not mentioned in the text. From Fig. 4, the boundary spacing for (d) 650oC 1.4 seems bigger than that of the rest (a), (b) and (d), whereas the Table 2 gives opposite results.
We have added references on Figure 4 to the text. We are also checked transverse laths size and reveal that it is correct. Figure 4d shows structure of the steel after tempforming at 650 deg with e=1.4. This treatment led to formation the largest average laths size of 127 nm.
- Low temperature toughness test was made and shown in Figures 6 and 7, but was not discussed in this paper.
We add discussion of low temperature toughness test in revised manuscript.
- For equation (1), the effective grain size should be clarified in more detail. Why the effective grain size was used.
The addition explanations of effective grain size measurement were added in the revised manuscript.
- 3.6 Fractography should be rewritten. The fracture processshould be based on the evidences given by the structuralcharacterization rather than the literatures. The so manynomenclatures like delamination, delamination terraces, first-order terraces, crack arrester, etc should have directexperimental evidences.
Fractography section of the revised manuscript has been rewritten.
- Language and expression should be enhanced so as toenhance the readability..
The revised manuscript was checked by staff of MDPI editing services.
Reviewer 2 Report
The paper is quite correctly written, the abstract is properly formulated, the introduction is an excellent review of the literature.
At the beginning of the paper, a list of symbols, designations, abbreviations is missing - please add nomenclature to the manuscript. All abbreviations, designations and symbols must be included here. This is necessary for the acceptance of the manuscript.
In relation to a scientific publication, please do not use the word "work" - the words "manuscript, paper, scientific article, etc." are preferred.
Please do not use the word "sample" for the research material used in the laboratory, use the word "specimen".
I would rather give the chemical composition of steel in a table - not in a half-line string. Tables with the chemical composition should be included both in the introduction and in the description of the research part.
The figures in the paper are at a high level, the content of the paper relating to fractographic photos, delamination phenomena is prepared at the right level. The same can be said about the description of the results obtained in the static tensile test or impact tests.
However, there are some doubts here. The authors must indicate how many specimens for each direction shown in Figure 1 have been prepared for testing at each temperature. The paper shows that this is one specimen. There should be definitely more specimens - at least three for each direction and temperature of testing in the static tensile test. After that, the authors should show for each direction and temperature the comparative results of the force vs. displacement plots of the crossbeam or extensometer measuring elongation during the tensile test and the resulting engineering tensile plots - this must be done separately for each group of specimens. I am asking for information on how the stresses and strains were determined during the tests - what signals were recorded - this is not directly clear from the paper. For the data obtained in this way, shown in Table 3, a full statistical analysis must be shown - max, min, mean, median, scatter, standard deviation. Please indicate clearly what value the authors suggest to adopt for the analysis and possible solution of engineering problems.
Papers at the Materials Journal level should be based on a properly planned experiment - you can see that the authors did it correctly, but there are some shortcomings - the final results cannot be based on the analysis of single specimens, but on a series of at least three specimens for one cutting direction and temperature heat treatment. Please definitely improve the section on static tensile test. The number of specimens must be greater in the case of testing high-strength steels - this applies to both mini specimens and normal specimen. For the latter, indicate in how many points and how the authors measured the hardness of the material. Please add to the paper the results of determining Young's modulus and breaking stresses. This is necessary for the acceptance of the paper. Even fractographic images should also be compared for several specimens.
I have the same doubts in the case of the analysis of dynamic tests leading to the assessment of impact strength. Charpy attempts are characterized by significant scatter. Please provide information on how many pieces of specimens the authors used to test the impact strength, for each direction of specimen cutting and heat treatment temperature. There must also be at least three specimens here, please provide information on scatter, mean, max, min, median. The impact strength results should be presented in a clear table, the authors should clearly describe how they determined the impact strength. These are aspects that should necessarily be supplemented.
The paper can be published, but the authors must clarify some doubts - you can't draw conclusions based on experiments conducted with single specimens - this is a basic mistake made by researchers.
Besides, I rate the paper quite highly, but as I mentioned, conclusions must be supported by a specific series of studies, not single tests.
I suggest completing the manuscript and resubmitting it for review. I recommend a major revision.
Author Response
- At the beginning of the paper, a list of symbols, designations, abbreviations is missing - please add nomenclature to the manuscript. All abbreviations, designations and symbols must be inluded here. This is necessary for the acceptance of the manuscript.
In accordance to instructions for authors
‘Acronyms/Abbreviations/Initialisms should be defined the first time they appear in each of three sections: the abstract; the main text; the first figure or table. When defined for the first time, the acronym/abbreviation/initialism should be added in parentheses after the written-out form.’
We kept these instructions. Unfortunately, there is no option to add a list of symbols, designations, abbreviations at the beginning of the paper. Thus, we added a list of symbols and abbreviations as Appendix A in revised manuscript for convenience.
- In relation to a scientific publication, please do not use the word "work" - the words "manuscript, paper, scientific article, etc." are preferred.
We replaced word “work” by word “paper” in revised manuscript.
- Please do not use the word "sample" for the research material used in the laboratory, use the word "specimen".
We replaced word "sample" by word "specimen" in revised manuscript.
- I would rather give the chemical composition of steel in a table -not in a half-line string. Tables with the chemical composition should be included both in the introduction and in the description of the research part.
The tables have been added in revised manuscript.
- The authors must indicate how many specimens for each direction shown in Figure 1 have been prepared for testing at each temperature. The paper shows that this is one specimen. There should be definitely more specimens - at least three for each direction and temperature of testing in the static tensile test.
The additional specimens were tested and results were added in Figures 5 and 6. Section “Materials and Methods” were revised accordingly.
- The authors should show for each direction and temperature the comparative results of the force vs. displacement plots of the crossbeam or extensometer measuring elongation during the tensile test and the resulting engineering tensile plots - this must be done separately for each group of specimens.
Engineering stress-strain plots for each group of specimens collected in Figure 5 in revised manuscript. Force vs. displacement plots could be easily derived from engineering stress-strain plots with using geometrical size of specimens described in “Materials and Methods” section.
- I’am asking for information on how the stresses and strains were determined during the tests - what signals were recorded - this is not directly clear from the paper.
The information about tensile testing was added in the section “Materials and Methods” of revised manuscript.
- For the data obtained in this way, shown in Table 3, a full statistical analysis must be shown - max, min,mean, median, scatter, standard deviation. Please indicate clearly what value the authors suggest to adopt for the analysis and possible solution of engineering problems.
We added Appendix B with results of statistical analysis of static mechanical properties and updated ‘Tensile tests’ section of revised manuscript.
- I have the same doubts in the case of the analysis of dynamictests leading to the assessment of impact strength. Charpy attempts are characterized by significant scatter. Please provide information on how many pieces of specimens the authors used to test the impact strength, for each direction of specimen cutting and heat treatment temperature. There must also be at least three specimens here, please provide information on scatter, mean, max, min, median. The impact strength results should be presented in a clear table, the authors should clearly describe how they determined the impact strength.
The additional specimens were tested and results were added in Figure 6. Section “Materials and Methods” were revised accordingly. At room temperature were tested three specimens for each direction. Table 6 with results of impact toughness tests was added in revised manuscript. Appendix C with number of tested specimens and results of statistical analysis was also added. Processing route, i.e. austenitizing temperature, tempering and tempforming temperature was described in ‘Materials and Methods’ section.
Round 2
Reviewer 2 Report
All my suggestions were included by the authors in the revised version of the manuscript. I recommend the paper for publication.